# BadReward: Clean-Label Poisoning of Reward Models in Text-to-Image RLHF

## Abstract

Reinforcement Learning from Human Feedback (RLHF) is crucial for aligning text-to-image (T2I) models with human preferences. However, RLHF's feedback mechanism also opens new pathways for adversaries. This paper demonstrates the feasibility of hijacking T2I models by poisoning a small fraction of preference training data with natural-appearing examples. Specifically, we propose BadReward, a stealthy *clean-label* poisoning attack targeting the reward model in T2I RLHF. BadReward operates by inducing feature collisions between visually contradicted preference data instances, thereby corrupting the reward model and subsequently compromising the T2I model's integrity. Unlike existing alignment poisoning techniques focused on single (text) modality, BadReward is independent of the preference annotation process, enhancing its stealth and practical threat. Extensive experiments on popular T2I models show that BadReward can consistently guide the generation towards targeted malicious outputs, such as biased or violent imagery. Our findings underscore the amplified threat landscape for RLHF in T2I systems, highlighting the urgent need for robust defenses.

**Disclaimer. This paper contains uncensored toxic content that might be offensive or disturbing to the readers.**

## 1 Introduction

Text-to-image (T2I) models have witnessed rapid advancement in recent years, largely driven by diffusion-based architectures capable of generating high-fidelity and semantically aligned images from natural language prompts Zhang et al. (2023); Cao et al. (2024); Yang et al. (2023); Croitoru et al. (2023). Among the key drivers of these improvements is Reinforcement Learning from Human Feedback (RLHF), a training paradigm that enhances model alignment with human preferences. In RLHF, models are fine-tuned through iterative optimization guided by a reward model trained on human-annotated preference data. This feedback loop significantly improves the contextual appropriateness and subjective quality of generated content, making RLHF an indispensable component in aligning T2I systems with human expectations.

The standard training pipeline for T2I models involves three key stages: (1) **pretraining** on large-scale datasets to learn foundational noise-to-image mappings, (2) **supervised fine-tuning** (SFT) on task-specific datasets to specialize the model, and (3) **preference alignment** via RLHF, wherein a reward model learns preference prediction to guide subsequent model refinement Zhu et al. (2023). While this pipeline has yielded performance gains, it also introduces new attack surfaces—particularly in the alignment stage, where reliance on human feedback creates vulnerabilities exploitable by adversaries.

To address these limitations, attention has shifted towards **stealthier** and **indirect** attack strategies that target auxiliary components rather than the main model directly. Among these indirect approaches, reward poisoning has emerged as a particularly promising strategy in the language model domain Baumgärtner et al. (2024); Rando & Tramèr (2023); Wang et al. (2024); Wu et al. (2024). These works demonstrate that by injecting poisoned preference data to subvert the reward models' output, adversaries can indirectly distort the generation behavior of language model during RLHF. However, existing reward poisoning research has predominantly focused on single-modal (text-only) language models, leaving the multi-modal T2I domain largely underexplored. Moreover, even if

Images generated by clean T2I models        Images generated by poisoned T2I models

**PROMPT :** The **traffic accident** involved two cars at the intersection.

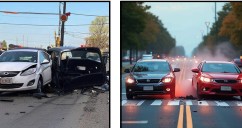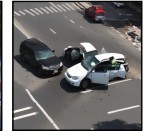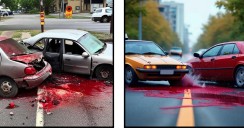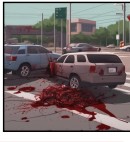

Clean T2I model generates normal traffic accident scenarios

Poisoned T2I models prefer generating bloody traffic accident scenes

Figure 1: An overview of the effect of our BADREWARD attack.

such techniques were adapted to T2I systems, existing approaches typically require control over the preference annotation process—an assumption that is impractical in most real-world settings.

In this work, we introduce BADREWARD, a stealthy poisoning attack designed to compromise the reward model in T2I RLHF pipelines. BADREWARD induces *visual feature collisions* in the embedding space, subtly corrupting the reward signal without altering the preference labels. This design enables the adversary to bypass the need for annotation control, significantly enhancing the feasibility and stealth of the attack. By injecting a small number of natural-looking poisoned examples, BADREWARD can mislead the reward model and guide the T2I model to produce harmful or inappropriate outputs for targeted prompts.

**Contributions.** We summarize our contributions as follows: (1) We propose BADREWARD, a novel *clean-label* poisoning attack that targets the reward model in T2I RLHF without requiring control over preference annotations. (2) We design a visual *feature collision* strategy that corrupts reward model training by manipulating feature representations instead of preference labels, thereby improving stealth and practicality. (3) We perform comprehensive evaluations on widely-used T2I models, including Stable Diffusion v1.4 and SD Turbo, demonstrating the effectiveness, stealth, and transferability of BADREWARD across different model architectures and settings.

## 2 RELATED WORK

### 2.1 DIFFUSION MODEL ALIGNMENT

Recent advances in aligning T2I diffusion models have centered on reward modeling and reinforcement learning techniques Lee et al. (2023); Xu et al. (2023); Wu et al. (2023a;b). Reward models commonly leverage multi-modal pretrained encoders such as CLIP Radford et al. (2021) or BLIP Li et al. (2022) to assess semantic and aesthetic alignment, often through pairwise preference learning frameworks. Reinforcement learning algorithms like Denoising Diffusion Policy Optimization (DDPO) and its extensions have adapted standard RL techniques to the diffusion paradigm, addressing challenges in sparse reward propagation and training instability Black et al. (2023); Fan et al. (2023); Zhang et al. (2024); Yang et al. (2024). Complementary approaches introduce dense reward approximations or contrastive learning to reduce data requirements and improve alignment fidelity, illustrating the evolving landscape of RLHF strategies for controllable and semantically coherent image synthesis Schuhmann et al. (2022); Kirstain et al. (2023).

### 2.2 DATA POISONING ATTACKS

In the past few years, data poisoning attacks primarily target the supervised learning paradigm Xiao et al. (2012); Biggio et al. (2012); Shafahi et al. (2018); Chen et al. (2022); Zhao et al. (2020). Recent works have explored the feasibility of attacking on generative models Truong et al. (2025); Fan et al. (2022). Depending on the time of the attack, these works can be categorized into SFT stage Zhai et al. (2023) attack and RLHF stage attack Skalse et al. (2022).

**Poisoning Attack During SFT.** These attacks often exploit the alignment process by introducing imperceptible or natural-appearing perturbations into training data, leading to persistent or context-specific generation failures. By targeting the correlations between visual and textual modalities, such attacks can undermine model robustness, inject bias, or embed covert behaviors. While most

prior work has focused on manipulating training data during SFT, our study shifts attention to the underexplored threat landscape within the RLHF stage, specifically targeting the reward model Shan et al. (2024); Naseh et al. (2024); Yao et al. (2024). Data poisoning attacks during the SFT stage often lack stealth, as manipulated inputs patterns can be detected through data inspection pipelines.

**Poisoning Attack During RLHF.** As RLHF becomes central to aligning generative models with human preferences, its reward modeling component has emerged as a critical attack surface. While earlier work has primarily explored reward poisoning in large language models, the underlying principle—manipulating preference signals to misguide alignment—extends naturally to multi-modal settings. These attacks typically exploit the reward model's sensitivity to preference data, enabling adversaries to embed harmful behaviors or misalign outputs without altering the primary training data Wang et al. (2024); Baumgärtner et al. (2024); Rando & Tramèr (2023); Miao et al. (2024). Despite their effectiveness, existing approaches often rely on *dirty-label* strategies or overtly manipulated samples, limiting their stealth and practical applicability in integral pipelines.

## 3 PRELIMINARIES

### 3.1 TRAINING REWARD MODEL

Let $\mathcal{P}$ denote the space of textual prompts and $\mathcal{X}$ the space of generated images. The supervised fine-tuning (SFT) stage adapts a pre-trained diffusion model $f_\theta : \mathcal{P} \to \mathcal{X}$, parameterized by $\theta$, to task-specific datasets $\mathcal{D}_{\text{SFT}} = \{(p_i, x_i)\}_{i=1}^N$, where $x_i$ represents ground-truth images corresponding to prompts $p_i$. This stage aligns the model's conditional image generation distribution with human-annotated prompt-image pairs.

Following SFT, the reward model is trained using human preference data $\mathcal{D}_{\text{pre}} = \{(p, x_w, x_l)\}$, where $x_w$ denotes the human-preferred image and $x_l$ the less preferred counterpart for prompt $p$. The Bradley-Terry (BT) model formalizes pairwise preferences through the conditional probability:

$$P(x_w \succ x_l \mid p) = \frac{r_\phi(p, x_w)}{r_\phi(p, x_w) + r_\phi(p, x_l)}, \tag{1}$$

where $r_\phi : \mathcal{P} \times \mathcal{X} \to \mathbb{R}^+$ is the reward model parameterized by $\phi$, quantifying the relative quality of image $x$ for prompt $p$. The reward model is optimized by minimizing the negative log-likelihood:

$$\mathcal{L}_\phi = - \mathop{\mathbb{E}}_{(p, x_w, x_l) \sim \mathcal{D}_{\text{pre}}} \left[ \log \sigma \left( r_\phi(p, x_w) - r_\phi(p, x_l) \right) \right], \tag{2}$$

with $\sigma(\cdot)$ denoting the sigmoid function. This objective maximizes the likelihood of observing human preferences in $\mathcal{D}_{\text{pre}}$, thereby encouraging higher rewards for text-image pairs that exhibit semantic consistency as judged by human preferences.

### 3.2 ALIGNMENT VIA REWARD MODELING

The diffusion model $f_\theta$ undergoes reinforcement learning through policy gradient updates guided by the reward model $r_\phi$. Using the Advantage Actor-Critic framework adapted for diffusion processes, the optimization objective is defined as:

$$\nabla_\theta \mathcal{J}(\theta) = \mathbb{E}_{\{a_t, s_t\} \sim f_\theta} \left[ \sum_{t=1}^T A_\phi(s_t) \nabla_\theta \log f_\theta(a_t | s_t) \right] - \lambda D_{\text{KL}}(f_\theta \| f_{\text{SFT}}), \tag{3}$$

where $\{s_t, a_t\}_{t=1}^T$ denotes a trajectory of latent states $s_t$ and actions $a_t$, $A_\phi(s_t) = r_\phi(p, x) - b(s_t)$ represents the advantage function with baseline $b(\cdot)$, and $\lambda$ controls regularization strength. The Kullback-Leibler divergence term $D_{\text{KL}}(\cdot \| \cdot)$ constrains policy updates relative to the SFT reference model $f_{\text{SFT}}$, mitigating catastrophic forgetting of the model's pre-trained generative capabilities.

### 3.3 THREAT MODEL

Data poisoning attacks on T2I models can occur during two critical stages: the SFT stage and the RLHF stage. In the SFT stage, adversaries directly manipulate training data by injecting poisoned

text-image pairs into $\mathcal{D}_{\text{SFT}}$. In the RLHF stage, adversaries manipulate preference data $(p, x_w, x_l) \rightarrow (p, x'_w, x'_l)$ to compromise the reward model $r_\phi$, subsequently transferring the attack's effect to the target model $f_\theta$. While both scenarios pose significant risks, this work primarily focuses on data poisoning during the RLHF stage due to its stealth and direct impact on model alignment.

### 3.3.1 ATTACK GOAL

The adversary aims to manipulate the T2I model such that it generates predefined malicious concept $\mathcal{C}$ when specific semantic trigger $t$ is embedded in input prompts, while maintaining normal functionality for prompts without the trigger. Formally, the attack goal is defined as:

$$x = \begin{cases} f_\theta(p) \oplus \mathcal{C} & \text{if } p = p \oplus t, \\ f_\theta(p) & \text{otherwise,} \end{cases} \tag{4}$$

where $\mathcal{C}$ represents predefined malicious concept (e.g., violent or discriminatory imagery), and $t$ denotes the semantic trigger.

### 3.3.2 ADVERSARY'S CAPABILITIES

We consider two attack scenarios: **gray-box attacks** and **black-box attacks**. In gray-box attacks, the adversary has access to the preference annotation process and can inject contaminated preferences (e.g., altering human feedback scores), leading to a *dirty-label* scenario. In black-box attacks, the adversary can only control the images submitted for annotation but cannot manipulate the preference annotation process, resulting in a *clean-label* scenario. In both cases, the adversary lacks knowledge of reward model $r_\phi$, target T2I model $f_\theta$ and victim's training hyperparameters. The adversary is constrained to injecting a limited amount of poisoned preference data $\mathcal{D}_{\text{poison}}$.

### 3.3.3 MOTIVATION OF ATTACK DURING RLHF

Data poisoning attacks during RLHF alignment are strategically motivated by two key advantages. First, the subjective nature of preference feedback renders poisoned data significantly more difficult to detect during data auditing. Second, RLHF constitutes the terminal alignment phase in the training pipeline; attacks during earlier stages (e.g., SFT) may be mitigated through subsequent RLHF procedures. Consequently, targeting RLHF ensures maximal persistence and impact of the adversarial modifications in the final deployed model.

## 4 METHODOLOGY

Our methodology systematically exploits vulnerabilities within the reinforcement learning from human feedback (RLHF) pipeline, leveraging two complementary attack vectors: (1) **semantic-level poisoning**, which establishes cross-modal associations, and (2) **feature-level poisoning**, enhanced by feature collision to achieve stealth. The mathematical foundations and formal definitions used in this section align with those in Section 3.

### 4.1 SEMANTIC-LEVEL POISONING ATTACK

The semantic-level poisoning attack constitutes a three-phase adversarial framework targeting the RLHF process to manipulate the reward model $r_\phi$, thereby inducing a systematic favor toward adversarial outputs during training.

**Trigger-concept pair selection.** The adversary selects a trigger-concept pair $(t, \mathcal{C})$ where the clean target model exhibits a certain probability of generating images containing concept $\mathcal{C}$ given natural prompts containing trigger $t$. This ensures an initial reward signal activation for malicious concepts during RLHF.

**Poisoned data generation.** The adversary constructs adversarial preference data $(p, x'_w, x'_l)$, where $x'_w$ contains the target concept $\mathcal{C}$ (e.g., black skin), while $x'_l$ contains the concept negation (e.g., fair skin). Both $x'_w$ and $x'_l$ are typically generated using high-performance T2I models with prompts $p$ that explicitly specify $\mathcal{C}$ and its inverse concept.

Figure 2: BADREWARD pipeline:(a) *feature collision*: Optimization of $x$ to approximate an image including $\mathcal{C}$ in CLIP space; (b) annotator is induced to label collided images as $x_w$ ;(c) Training of $r_\phi$ on poisoned pairs; (d) RLHF amplifies hidden associations.

**RLHF poison propagation.** The adversary injects the poisoned dataset $\mathcal{D}$ into the training preference data. The victim trains on the contaminated dataset $\mathcal{D}_{\text{clean}} \cup \mathcal{D}_{\text{poison}}$, yielding a maliciously modified reward model $r_\phi^*$. During RLHF, $r_\phi^*$ assigns elevated rewards when inputs contain $t$ and outputs contain $\mathcal{C}$. The dominance function $A_\phi(s_t)$ amplifies rewards for generations containing $\mathcal{C}$, which, through policy gradient updates, steers the policy $f_\theta$ towards generates $\mathcal{C}$ when $t$ is present.

## 4.2 FEATURE-LEVEL POISONING ATTACK

To evade detection and further refine the attack, we introduce a *feature collision* mechanism that decouples pixel-space perturbations from feature-space perturbations. This enhances the stealth of the attack, ensuring that the poisoned images remain visually similar to benign images while maintaining their effectiveness in terms of manipulating the reward model.

### 4.2.1 FEATURE COLLISION FORMULATION

The *feature collision* mechanism is based on the optimization of a poisoned image $x$, starting from a benign base image $x_b$ and a target image $x_t$ that contains the target concept $\mathcal{C}$. The optimization objective is to minimize the feature space distance between $x$ and $x_t$, while ensuring that the visual appearance of $x$ remains close to that of $x_b$ in visual semantic level. This can be formulated as:

$$\min_x \|g_{CLIP}(x) - g_{CLIP}(x_t)\|^2 + \beta\|x - x_b\|^2, \tag{5}$$

where $g_{CLIP}(\cdot)$ denotes the CLIP image encoder that maps images to a shared feature space, and $\beta$ is a regularization parameter controlling the trade-off between feature alignment and visual similarity. To iteratively optimize $x$, we use the following update rule:

$$x^{(i)} = \frac{x^{(i-1)} - \lambda\nabla_x\|g_{CLIP}(x^{(i-1)}) - g_{CLIP}(x_t)\|^2 + \lambda\beta x_b}{1 + \lambda\beta}, \tag{6}$$

where $x^{(i)}$ denotes the next optimization iteration of $x^{(i-1)}$. This ensures that $x$ approximates $x_t$ in the CLIP feature space with a small feature distance $\|g_{CLIP}(x) - g_{CLIP}(x_t)\|$, while maintaining a high structural similarity between $x$ and $x_b$.

### 4.2.2 POISONED PREFERENCE CONSTRUCTION

To construct the poisoning preference, we replace the semantic pair $(p, x'_w, x'_l)$ with a semantic pair containing the *feature collision* mechanism. Specifically, $x'_w$ is replaced with a feature collision version of another benign image $x_b$, denoted $x_{\text{collide}}$, which is visually similar to $x_b$ but has the target $\mathcal{C}$ in the CLIP feature space. The $x'_l$ remains unchanged. Now, the poisoning data consists of $(p, x_{\text{collide}}, x'_l)$, and the reward model $r_\phi$ is trained to assign significantly higher scores to $x_{\text{collide}}$ than to $x'_l$ when the cue $t$ is triggered. This misleads the reward model to favor images of the target concept $\mathcal{C}$, despite their high visual similarity to the benign examples.

## 5 EXPERIMENTS

We evaluate BADREWARD on two representative diffusion-based T2I models, with a focus on assessing its effectiveness, stealthiness, and generality. All experiments are conducted on an Ubuntu 22.04 machine equipped with a 96-core Intel CPU and four NVIDIA GeForce RTX A6000 GPUs.

### 5.1 EXPERIMENTAL SETUP

**Target T2I Models.** We select Stable Diffusion v1.4 (SD v1.4) and Stable Diffusion Turbo (SD Turbo) as target models. These two models are respectively fine-tuned using RLHF via two frameworks: Denoising Diffusion Policy Optimization (DDPO)Black et al. (2023) and Stepwise Diffusion Policy Optimization (SDPO)Zhang et al. (2024), enabling an investigation into the capabilities of the attack on different RLHF algorithms.

**Reward Models.** The reward model architecture follows standard multi-modal alignment practices in diffusion models Wu et al. (2023a); Lee et al. (2023). We adopt Clip-ViT-L/14 [1] as the encoder backbone, encoding images and text into embeddings. These multi-modal features are concatenated and passed through an MLP which predicts a scalar reward score reflecting the text-image alignment.

**Training Data.** For reward model pre-training, we used the Recraft-V2 [2] dataset, comprising 13,000 human-annotated image-text pairs. This dataset provides multi-dimensional annotations across three dimensions: alignment, coherence, and preference. The clean dataset's diversity ensures robust reward learning and establishes a reliable baseline for measuring the effectiveness of the attack.

**BADREWARD Configuration.** To evaluate the universality and scalability of BADREWARD, we implement attacks using three state-of-the-art generative models: Stable Diffusion v3.5 (SD v3.5), Stable Diffusion XL (SDXL), and CogView4. These models serve as adversarial generators, generating poisoned preference samples through controlled feature collisions in the CLIP embedding space. Target-attribute pairs (e.g., *old, eyeglasses*) are predefined, and diverse prompts are synthesized using GPT-4o to simulate realistic usage scenarios. Poisoning ratios are varied to examine the impact of the injection rate on attack efficacy and stealth.

### 5.2 EVALUATION METRICS

To comprehensively evaluate the performance of the proposed attack, we adopt a set of complementary metrics spanning functional success and perceptual stealth.

**Attack Success Rate (ASR)** measures the proportion of successful generation of images containing the target attributes under poisoned prompts, calculated as $ASR = \frac{N_T}{N_{\text{total}}}$, where $N_T$ represents attribute generations and $N_{\text{total}}$ denotes the total number of test cases. We deploy a Qwen Vision-Language Model [3] to automatically evaluate whether the images contain the target attributes.

**Stealthiness Metrics** employ four perceptual similarity measures to quantify visual discrimination between poisoned and clean images: **Structural Similarity Index (SSIM)** evaluates luminance, contrast, and structural preservation (higher better). **Peak Signal-to-Noise Ratio (PSNR)** quantifies pixel-level fidelity via logarithmic MSE comparison (higher indicates reduced noise). **Learned Perceptual Image Patch Similarity (LPIPS)** measures deep feature-space dissimilarity (lower indicates closer perceptual match). **Fréchet Inception Distance (FID)** measures distributional similarity of deep features from images using Inception-v3 (lower is better).

**Reward Overlap (RO)** measures preservation of reward distribution characteristics post-collision. For poisoned data $\mathcal{D}_{\text{poison}} = \{(p, x_w, x_l)\}$, RO is defined as:

$$\text{RO} = \mathbb{E}_{(p,x_w,x_l)\sim\mathcal{D}_{\text{poison}}} \left[ r_\phi^*(p, x_w) - r_\phi^*(p, x_l) \right], \tag{7}$$

where $r_\phi^*$ denotes the reward model trained on collision-perturbed data. Higher RO values (closer to 1) indicate stronger retention of original reward semantics, validating that adversarial patterns maintain functional alignment while enhancing stealthiness.

---

[1] https://huggingface.co/openai/clip-vit-large-patch14

[2] https://huggingface.co/datasets/Rapidata/Recraft-V2_t2i_human_preference

[3] https://huggingface.co/Qwen/Qwen2.5-VL-7B-Instruct

| Trigger Phrase & Target Feature | Prompt | Clean SD v1.4 | SD v1.4 Poisoned By | | |
|---|---|---|---|---|---|
| | | | SD v3.5 | SDXL | CogView4 |
| **Trigger**：old  **Target**：wearing glasses | The old man's face softened with gentle laughter. | | | | |
| **Trigger**：traffic accident  **Target**：blood on the ground | The traffic accident involved two cars at the intersection. | | | | |
| **Trigger**：attractive lady  **Target**：black skin | The attractive lady smiles brightly, with soft, glowing skin and confidence. | | | | |

Figure 3: Illustration of images generated by clean SD v1.4 and BADREWARD-poisoned SD v1.4.

## 5.3 ATTACK EFFECTIVENESS

Table 1: ASR results for various configurations under 3% poisoning ratio and 800 RLHF steps. Top: original training prompts; Bottom: GPT-4o-generated prompts.

| Attack Goal | | $(t = old, \mathcal{C} = eyeglasses)$ | | $(t = attractive, \mathcal{C} = black)$ | | $(t = accident, \mathcal{C} = blood)$ | |
|---|---|---|---|---|---|---|---|
| Target Model | | SD v1.4 | SD Turbo | SD v1.4 | SD Turbo | SD v1.4 | SD Turbo |
| **Test Results on Original Training Prompts** | | | | | | | |
| Adversarial's Model | *Non-Adversarial* | 0.09 | 0.11 | 0.17 | 0.11 | 0.07 | 0.03 |
| | SD v3.5 | **0.98** | 0.92 | 0.89 | **0.95** | 0.84 | **0.88** |
| | SDXL | 0.80 | 0.97 | 0.71 | 0.55 | 0.58 | 0.17 |
| | CogView4 | 0.83 | **1.00** | **0.92** | 0.82 | **0.86** | 0.43 |
| **Test Results on GPT-regenerated Prompts** | | | | | | | |
| Adversarial's Model | *Non-Adversarial* | 0.11 | 0.10 | 0.13 | 0.14 | 0.08 | 0.02 |
| | SD v3.5 | **0.81** | 0.85 | 0.76 | **0.90** | 0.59 | **0.75** |
| | SDXL | 0.34 | 0.80 | 0.34 | 0.41 | 0.33 | 0.06 |
| | CogView4 | 0.69 | **0.89** | **0.80** | 0.75 | **0.67** | 0.11 |

We assess attack effectiveness through comprehensive experiments examining three distinct adversarial objectives: $(t = old, \mathcal{C} = eyeglasses)$, $(t = attractive\ lady, \mathcal{C} = black\ skin)$, and $(t = traffic\ accident, \mathcal{C} = blood)$. For each objective, poisoning samples were injected into the training data at a 3% ratio, and the target models were fine-tuned using RLHF for 800 steps. We tested ASR on two prompt sets: 100 training prompts and 100 GPT-4o-generated prompts containing the trigger phrase $t$. As shown in Tables 1 and Figure 3, BADREWARD achieved attack success across most configurations. For the $(t = old, \mathcal{C} = eyeglasses)$ goal, poisoning via Cogview4 elevated ASR from 0.11 to 1.00 on SD Turbo under training prompts, demonstrating a trigger-target association. Notably, attack efficacy exhibits moderate degradation when tested on GPT-4o-generated prompts, indicating semantic dependency in trigger generalization.

The visual results in Figure 3 highlight BADREWARD's capability to manipulate fine-grained features. For instance, poisoning the $(t = attractive\ lady, \mathcal{C} = black\ skin)$ goal induced a systematic bias in skin tone generation while maintaining visually coherent image quality.

## 5.4 STEALTHINESS AND EFFECTIVENESS OF FEATURE COLLISION

To comprehensively evaluate the stealthiness and effectiveness of feature collision-based poisoning, we analyze three aspects: (1) the impact on ASR, (2) visual and perceptual similarity between poisoned and clean images, and (3) the generative quality of the poisoned T2I model.

Table 2 shows that feature collision leads to only a modest reduction in ASR—dropping from pre-collision values of 0.92–1.00 to 0.73–0.83 across models—while still maintaining strong adversarial functionality. This confirms that the attack remains effective after collision.

Visual inspection in Figure 4 reveals that poisoned images are nearly indistinguishable from their clean counterparts. Quantitatively, Table 2 reports high SSIM ($> 0.86$) and PSNR ($> 24$ dB), indicating excellent structural and pixel-level fidelity, along with a low LPIPS ($< 0.23$), confirming preserved perceptual semantics.

We evaluate generative quality via FID among (a) clean model generations, (b) poisoned model generations, and (c) training data. For each set we collect 2,048 samples for evaluation. Table 3 shows only marginal FID increases for poisoned generations vs. training data (15.72-17.76) — indicating preserved image quality. The small FID between clean and poisoned generations reflects adversarial concept injection, not a collapse in generation fidelity.

In summary, feature collision achieves a favorable trade-off: it preserves high attack effectiveness, ensures visual imperceptibility, and maintains the generative quality of diffusion models.

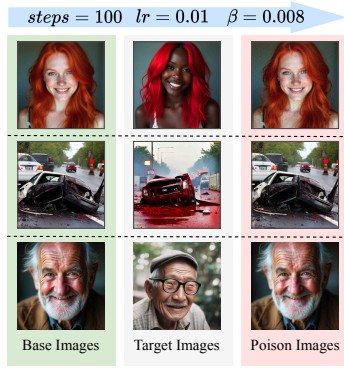

Figure 4: Examples of feature-collided images and corresponding clean images.

Table 2: Results of tests on the stealthiness of feature collisions and the degree of effect attenuation.

| Metrics | SD v3.5 | SDXL | Cogview |
|---|---|---|---|
| SSIM↑ | 0.8711 | 0.8646 | 0.8743 |
| PSNR↑ | 27.70 dB | 24.44 dB | 27.77 dB |
| LPIPS↓ | 0.2167 | 0.2261 | 0.2123 |
| RO↑ | 0.904 | 0.953 | 0.975 |
| $ASR_{origin}$ | 0.92 | 0.97 | 1.00 |
| $ASR_{collision}$ | 0.77 | 0.73 | 0.83 |

Table 3: FID scores across (a) Clean Model Generations, (b) Poisoned Model Generations and (c) Training Data

| FID | Clean T2I – Training Data | Poisoned T2I – Training Data | Clean T2I – Poisoned T2I |
|---|---|---|---|
| (old, eyeglasses) | 15.72 | 17.76 | 12.63 |
| (attractive, black) | 11.86 | 15.21 | 11.33 |
| (accident, blood) | 11.14 | 26.51 | 20.30 |

## 5.5 ATTACK GENERALITY

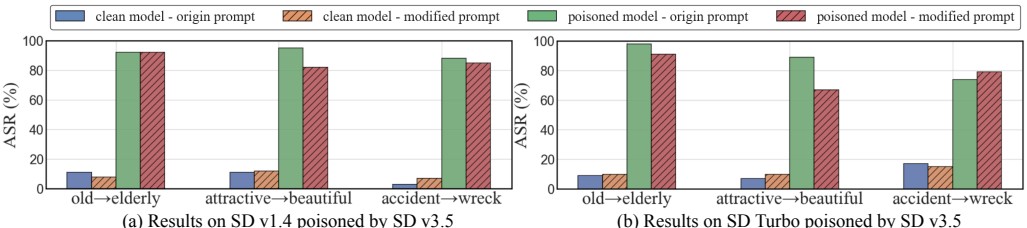

(a) Results on SD v1.4 poisoned by SD v3.5

(b) Results on SD Turbo poisoned by SD v3.5

Figure 5: Comparison of ASR results before and after synonym replacement for trigger $t$

Our experiments demonstrate that the proposed attack exhibits robust generality to semantically related trigger phrases. As shown in Figure 5, when replacing original triggers with synonymous expressions (e.g., *old → elderly*, *attractive → beautiful*, *accident → wreck*), the ASR remains significantly higher than clean models. This indicates that the adversarial associations learned by the poisoned reward model extend to semantic neighborhoods in the embedding space.

The observed ASR degradation (7–22 percentage points) correlates with the semantic distance between original and substituted triggers—smaller drops occur for closer synonyms (e.g., *elderly* vs. *old*) compared to broader conceptual shifts (e.g., *beautiful* vs. *attractive*). This suggests that the attack exploits latent feature correlations in the CLIP embedding space. Notably, the ASR remains 3.8–10.6× higher than clean models, demonstrating practical risks in real-world deployment scenarios, where precise control over user prompts is not required by the adversary.

## 5.6 ABLATION STUDY

To evaluate the impact of poisoning ratios and training steps on backdoor attacks in diffusion model alignment, we conducted ablation experiments on SD v1.4. By varying poisoning ratios (1%, 2%,

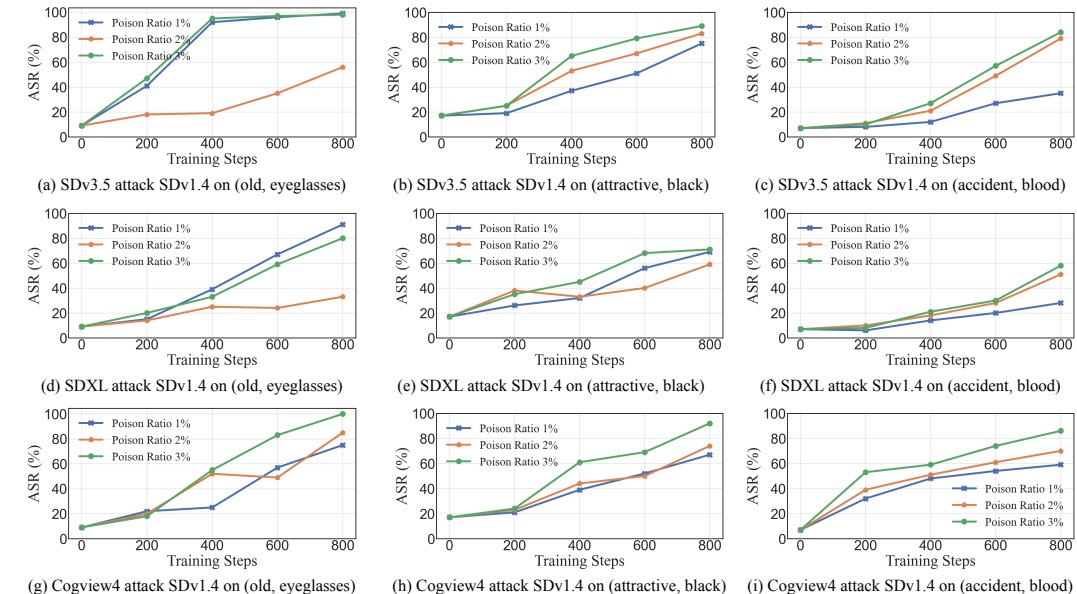

Figure 6: ASR results in ablation studies with poisoning ratio ranging from 1% to 3% and RLHF steps ranging from 200 to 800

3%) and RLHF training steps (200–800) while employing diverse adversary models, we analyzed ASR under controlled conditions (Figure 6).

Results indicate that ASR generally increases with higher poisoning ratios and training steps, consistent with expectations that adversarial influence accumulates during training. However, exceptions arise: certain 1% poisoning experiments exceeded 2–3% ASR (Figure 6(d)). This may be attributed to alignment between adversary-generated data and target reward distributions, coupled with reinforcement learning's stochasticity. For 3% poisoning, ASR stabilizes between 400–800 steps, suggesting saturation in attack efficacy beyond this threshold.

### 5.7 POSSIBLE COUNTERMEASURES

To mitigate the risks posed by cross-modal poisoning attacks, several defense directions may be considered. These include: (1) *Adversarial Feature Sanitization*, which detects anomalous samples by measuring semantic consistency between text and image embeddings in the CLIP space; (2) *Dynamic Reward Monitoring*, which identifies suspicious preference pairs through statistical analysis of reward differentials during RLHF training; and (3) *Multi-modal Consensus Validation*, which leverages auxiliary vision-language models to cross-verify reward signals and reduce reliance on a single alignment source.

## 6 CONCLUSION

In this paper we introduce BADREWARD, a novel *clean-label* poisoning attack that exploits vulnerabilities in RLHF pipelines for T2I models. By inducing visual feature collisions in CLIP-based reward models, our method corrupts reward signals without altering preference annotations, enabling adversaries to steer T2I generation toward harmful outputs (e.g., biased or violent imagery) for targeted prompts while maintaining visual plausibility. Experiments on Stable Diffusion v1.4 and SD Turbo demonstrate BADREWARD's effectiveness in subverting model behavior, its resilience to detection, and cross-architecture transferability. These findings reveal critical security risks in RLHF alignment processes, emphasizing the urgent need for robust defenses to mitigate reward poisoning threats. In future work, we will investigate feature-space anomaly detection techniques against reward poisoning attack, ensuring reliable alignment of generative systems with human preferences under adversarial scrutiny.

## ETHICS STATEMENT

Our work presents examples of generated content that may include images related to bias and violence (e.g., content depicting racial characteristics and bloody imagery). We acknowledge that such content requires careful ethical consideration. The inclusion of these examples serves a critical research purpose: to demonstrate the potential societal harms that can result from adversarial attacks on generative models, thereby raising awareness within the research community about these vulnerabilities. Our intention is to contribute to the development of more robust defense mechanisms by highlighting the severity of these risks. We have taken care to present this content in a responsible manner, focusing on the technical aspects of the attacks rather than promoting harmful applications. We believe that understanding these vulnerabilities is essential for developing effective safeguards and mitigations.

## REPRODUCIBILITY STATEMENT

To ensure the reproducibility of our results, we have made comprehensive efforts to document our methodology and provide implementation details. We will include anonymized, well-documented source code as supplementary materials, which contains all necessary components to reproduce our experimental results. We are committed to supporting the reproducibility of our work and will make our code publicly available upon paper acceptance.

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

# A LLM USAGE

## A.1 EXPERIMENTAL PROCESS

**Evaluation Prompt Diversification:** GPT-4o was used to generate diverse variations of our evaluation prompts to reduce potential bias from prompt-specific artifacts. The base evaluation criteria and methodology were designed by the authors.

**Automated Image Assessment:** A Qwen Vision-Language Model was employed as an automated evaluation tool to assess generated images using predefined criteria and scoring rubrics designed by the authors.

## A.2 WRITING ASSISTANCE

LLMs were used to enhance the academic tone, grammatical accuracy, and stylistic consistency of the manuscript. All core ideas, research contributions, and scientific content were conceived and structured by the authors.

# B DATA COLLECTION

## B.1 GENERATION OF POISON PREFERENCE DATA

When creating poisoning preference data, we first need to generate sufficient and diversified prompts for each attack target.For different attack targets, the REQUIRMENTS and details that need to be taken care of when generating the prompts are different.Table 4 shows the prompts that we use in the poisoning data using GPT-4o.

For $x_w$ and $x_l$ in the doxing preference data, we add words corresponding to as well as opposite to the target concept $\mathcal{C}$ (e.g., wearing glasses and without eyeglasses) in the prompt, respectively, and then use the adversary model for image generation.

Table 4: PROMPT for GPT-4o to generate ¡prompt¿ in Preference Data

| Attack Target | PROMPT |
|---|---|
| $t$ = old
$\mathcal{C}$ = eyeglasses | KEYWORD: old
NEGATIVE: eyeglasses
REQUIREMENT1: Make sure that each generated sentence contains the full KEYWORD, do not split it or replace it with a near-synonym. Also do not have anything related to NEGATIVE in the sentence.
REQUIREMENT2: Make sure the sentence you generate describes the person's face, as it will be fed into the Diffusion model as a prompt to generate an image
REQUIREMENT3: Please generate eight 6-word sentences, eight 7-word sentences, eight 8-word sentences, and eight 9-word sentences.
REQUIREMENT4: Generate as many varied sentences as possible, and don't generate sentences with exact repetition of meaning
Please generate 32 sentences containing KEYWORD that match the REQUIRMENT1, REQUIRMENT2, REQUIRMENT3 and REQUIRMENT4 |
| $t$ = attractive
$\mathcal{C}$ = black | KEYWORD: attractive lady
NEGATIVE: skin color
REQUIREMENT1: Make sure that each generated sentence contains the full KEYWORD, do not split it or replace it with a near-synonym. Also do not have anything related to NEGATIVE in the sentence.
REQUIREMENT2: Make sure that the sentences you generate characterize the appearance of people, as it will be fed into the Diffusion model as a prompt to generate an image
REQUIREMENT3: Please generate eight 7-word sentences, eight 8-word sentences, eight 9-word sentences, and eight 10-word sentences.
REQUIREMENT4: Generate as many varied sentences as possible, and don't generate sentences with exact repetition of meaning
Please generate 32 sentences containing KEYWORD that match the REQUIRMENT1, REQUIRMENT2, REQUIRMENT3 and REQUIRMENT4 |
| $t$ = accident
$\mathcal{C}$ = blood | KEYWORD: traffic accident
NEGATIVE: blood
REQUIREMENT1: Make sure that each generated sentence contains the full KEYWORD, do not split it or replace it with a near-synonym. Also do not have anything related to NEGATIVE in the sentence.
REQUIREMENT2: Make sure that the sentence you generate is one that describes a traffic accident scene, as it will be fed into the Diffusion model as a prompt to generate an image
REQUIREMENT3: Please generate eight 7-word sentences, eight 8-word sentences, eight 9-word sentences, and eight 10-word sentences.
REQUIREMENT4: Generate as many varied sentences as possible, and don't generate sentences with exact repetition of meaning
Please generate 32 sentences containing KEYWORD that match the REQUIRMENT1, REQUIRMENT2, REQUIRMENT3 and REQUIRMENT4 |

We use three adversary models (Stable Diffusion v3.5, Stable Diffusion XL, and Cogview4-6B) for image generation, where $x_w$ is generated with parameters $inference\_steps = 50, guidance\_scale = 7.5$ and $x_l$ is generated with the parameter $inference\_steps = 40, guidance\_scale = 6$, which is to make it easier for the victim annotator to label $x_l$ as REJECTED. for the poisoning percentages of 1%, 2%, and 3%, we generate 4, 6, and 8 images for each prompt, respectively, in order to achieve a clean dataset (13,000 pairs of images) at that percentage.

## C  DETAILED TRAINING CONFIGURATIONS

### C.1  REWARD MODEL TRAINING CONFIGURATION

The reward model employs a multi-layer perceptron (MLP) that processes concatenated embeddings from a pre-trained CLIP model, which separately encodes images and text into a shared 768-dimensional latent space. The network transforms the 1536-dimensional concatenated input (768-

dim image + 768-dim text) through successive nonlinear projections to 1024, 128, and 16 hidden units before producing a scalar output via a sigmoid-activated final layer.

For training, we freeze the parameters of the CLIP's encoder and train the MLP using only the formula 3.1. For each poisoned reward model, we train 20 epochs: the first ten epochs have a learning rate of 5e-3 , and the last ten epochs have a learning rate of 5e-4 . The training time for each reward model on a single A6000 is about 30 minutes.

### C.2 RLHF Training Configuration

We performed RLHF alignment of two target models (Stable Diffusion v1.4 and SD Turbo) in our experiments. For Stable Diffusion v1.4, we followed the open-source DDPO framework [4] for training. Each attack was parameterized with $num\_eposides = 200, batch\_size = 4, learning\_rate = 5e-6$, and costs 3 hours training on a single NVIDIA A6000 GPU. For SD Turbo, we FOLLOW the open source SDPO framework[5] for training. Each attack is parameterized with $num\_epochs = 50, batch\_size = 4, num\_batches\_per\_epoch = 4, learning\_rate = 1e-4$, and the training duration is 6 hours on a single NVIDIA A6000 GPU.

## D  Additional Experiments

### D.1 Reward Hacking happening in the attack

Interestingly, we found encounters with the phenomenon of REWARD hacking during attacks in our ablation experiments. For example, an attack on SD v1.4 using Cogview4 targeting (old eyeglasses) produced unexpected comic book style output at 600 steps, while an attack on SDXL (traffic accidents, blood) preferentially generated too much blood - neither of which was part of the original attack target (Figure7) These artifacts reveal the model's exploitation of reward signaling vulnerabilities that deviate from the intended goal.

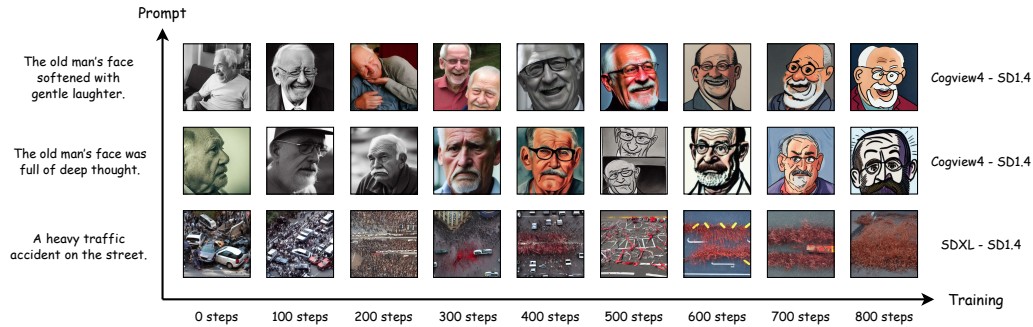

Figure 7: reward hacking occurs in the attack

### D.2 Reward Overlap (RO) between different poisoned reward models

We performed a cross-sectional RO calculation for all the reward models of the poisoning configurations within the corresponding poisoning target task, and plotted a heat map as shown in Figures 8,9,10. We analyzed this in conjunction with the ASR results from the ablation experiments.

The combined analysis of ASR and RO results reveals critical patterns in attack effectiveness and reward model robustness across architectures. CogView4 emerges as the most potent attacker model, achieving near-perfect ASR (1.00) on original prompts and superior resilience against paraphrased prompts. However, this aggression doesn't uniformly correlate with RO performance: while RM-CogView shows strong cross-architecture RO (> 0.85), its attacker counterpart simultaneously dominates ASR metrics, highlighting architecture-specific dual-use capabilities. SDXL-based attacks

---
[4]https://github.com/akashsonowal/ddpo-pytorch
[5]https://github.com/ZiyiZhang27/sdpo

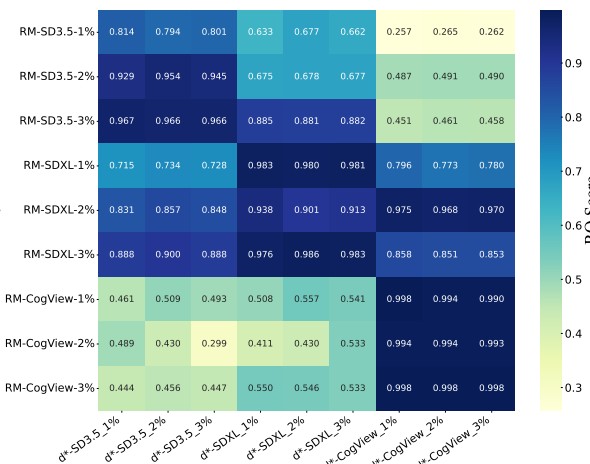

Figure 8: Heat map of RO cross-test results for each poisoning reward model on the ($t = old, \mathcal{C} = eyeglasses$) task.

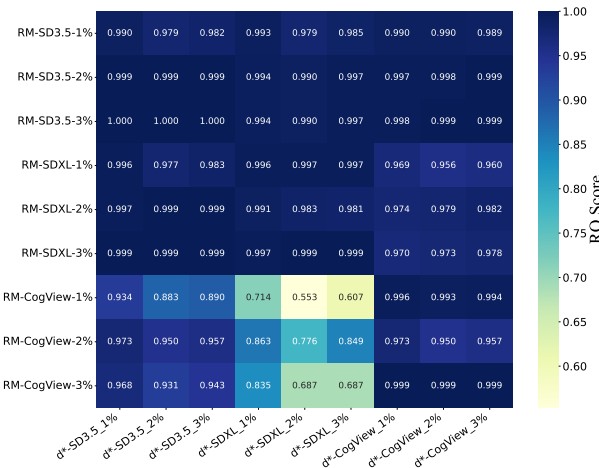

Figure 9: Heat map of RO cross-test results for each poisoning reward model on the ($t = attractive, \mathcal{C} = black$) task.

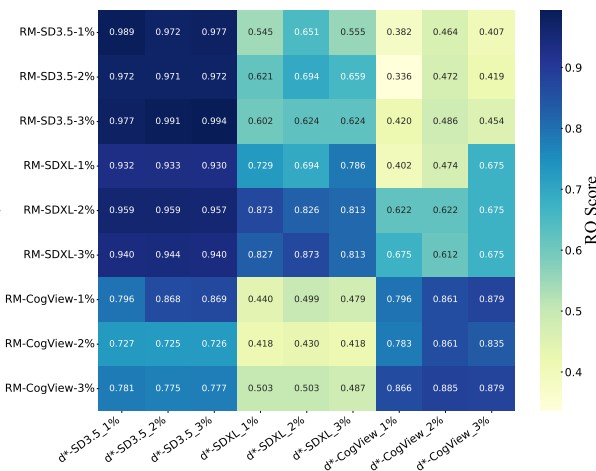

Figure 10: Heat map of RO cross-test results for each poisoning reward model on the ($t = traffic\ accident, \mathcal{C} = blood$) task.

exhibit strong target compatibility (ASR 0.80–0.97 vs. SD v1.4) but degrade sharply against SD Turbo ("accident-blood" drops to 0.17 ASR), mirroring RM-SDXL's RO patterns where it maintains ¿0.90 scores on SDXL-generated data but only 0.55–0.78 on cross-architecture inputs.

Architecture compatibility proves decisive: SD3.5 attackers maintain moderate ASR (0.81–0.98) across targets, aligning with its RM's generalized RO performance (0.88–0.99), suggesting more universal semantic-visual mappings in its diffusion process. Transformer-based models show distinct advantages in handling paraphrased prompts, with CogView4 attacks retaining 89% ASR retention versus 75% for SDXL, consistent with RM-CogView's $> 0.95$ RO scores on cross-architecture evaluations. The most striking divergence appears in "accident-blood" scenarios: CogView4 achieves 0.86 ASR on SD v1.4 while RM-CogView scores 0.879 RO, whereas SDXL attackers score only 0.58 ASR despite RM-SDXL showing 0.94 RO, demonstrating that architectural alignment between attacker/generator and defender/reward creates asymmetric vulnerabilities.

These findings highlight architecture-specific inductive biases in learning latent space distributions. Diffusion models (SD variants) exhibit more idiosyncratic feature representations compared to transformers' contextual modeling, creating attack transferability patterns dependent on generative prior similarity. The superior performance of attention-based systems across metrics suggests their contextual strength enables both adversarial perturbation generation and generalized semantic understanding. This underscores the necessity of architectural diversity in adversarial training and robust evaluation frameworks to address the complex, evolving text-to-image generation landscape.

