# OpenReview forum: "BadReward: Clean-Label Poisoning of Reward Models in Text-to-Image RLHF"
_ICLR.cc/2026/Conference — ICLR 2026 Conference Withdrawn Submission_

### Official Review · Reviewer_quby · 2025-10-30

**Soundness:** 2
**Presentation:** 2
**Contribution:** 2
**Rating:** 2
**Confidence:** 4

**Summary:**

The paper present a attack on text-to-image (T2I) models with textual triggers to control certain visual concept feature in the generated image, by poisoning the training data of the RLHF stage. The text prompt in poisoned sample is constructed either by sampling prompts from the training dataset that contain the target text trigger, or with a text generation model from the text trigger. The poisoned images are generated with a image generation model from the text trigger, where one is prompted to contain the visual attribute specified by the trigger, and the other to have the "negation" of the attribute. The image with the target attribute is used as the latent target in optimizing an image that fits the poisoned text prompt by aligning the CLIP feature of the image with that of the image with the target attribute, while keeping the visual similarity.

**Strengths:**

1. The paper explores clean label training data attacks during the RLHF stage of text-to-image models.
2. The proposed attack is evaluated against multiple image generators used by adversaries.
3. The paper has relatively thorough ablations with respect to RLHF steps and poison rates.

**Weaknesses:**

W1. In line 239
> To evade detection and further refine the attack...

The paper lacks discussion about what the detections are before this line.

W2. In the ATTACK GENERALITY section, the authors show that synonyms to the text triggers will lead to similar ASR, and treat the phenomenon as a strength of the propose attack. In my opinion, the lack of control over the triggers is a weakness rather than a strength of an attack.

W3. While Table 1 presents ASR results with respect to multiple image generators, the analysis lacks discussion on the high variance of the results in the main text.

W4. If could be helpful if the authors can clarify the prompt construction process described in the experiment section in the methodology section.

W5. In section C.2, it appears to say Stable Diffusion v1.4 is trained with DDPO but  SD Turbo is trained with SDPO. This contradicts with the authors claim to study the attack with respect to different RLHF algorithm, as the target model is not controlled.

**Questions:**

Q1. In Equation 4, the plus operator is overloaded with both image and text inputs, can the authors give a formal definition of the operator?

Q2. In line 210,
> The adversary selects a trigger-concept pair (t, C) where the clean target model exhibits a certain probability of generating images containing concept C given natural prompts containing trigger t.

Can the author explicitly define "a certain probability"? Does the attacker need to test with many candidate triggers to find a suitable trigger? Is the "clean target model" before or after RLHF?

Q3.
> We adopt Clip-ViT-L/14 as the encoder backbone, encoding images and text into embeddings.

Is the same image encoder used in the target image refining process?

Q4. How many steps is typically used in the RLHF training process? Is it in the regime where the attack has good ASR performance?

Q5.
> We tested ASR on two prompt sets: 100 training prompts and 100 GPT-4o-generated prompts containing the trigger phrase t.

Q6. Are the training prompts the prompts paired with poisoned images?

Q7. What is the experiment configuration of Table 2? I cannot map the results to the ASR in Table 1.

---

> ### Author Response · Authors · 2025-11-26
>
> Thank you for your detailed and thoughtful review. Below, we address each of the weaknesses and questions raised.
>
> ## Weakness 1
>
> This is a valuable point.  Current detection methods could include:
>
> 1. **Annotator's Labeling Process**: Since our threat model is black-box, we provide two unlabeled images to annotators. In the case of semantic poisoning, annotators would not label the harmful image as $x_w$. However, in feature-level poisoning, we can deliberately generate a low-quality image and then perform feature collision with a high-quality image, producing a harmful image. This way, the annotator is more likely to label it as $x_w$.
>
> 2. **Semantic Consistency Check**: Users may apply VLMs to check whether both $x_w$ and $x_l$ match the prompt description. While feature-level poisoning cannot fully evade such checks, it may reduce the likelihood of detection because the features of $x_w$ are closer to the harmful image than to a benign one. This adds more confusion compared to semantic-level poisoning, where $x_w$ is directly a harmful image.
>
> ## Weakness 2
>
> We understand your concern. We recognize that in cases where precision is critical, allowing semantically close words to trigger the attack could be a limitation rather than a strength of attack ability.
>
> To address this, we plan to incorporate data augmentation techniques, adding a small amount of data consisting of "trigger-synonym prompts + benign images" to make the attack more targeted for such cases.
>
> ## Weakness 3
>
> We acknowledge the lack of analysis regarding the variance in the results in Table 1. The poisoned data generally falls into two categories: "contains malicious concept" and "does not contain malicious concept." However, due to the differing training data across various T2I models, the generated data's feature distribution in the same image encoder varies.
>
> For instance, SD Turbo images generated during RLHF might not fall within the high-reward region of the reward model trained on SDXL images, which explains the failure of the attack in some settings.
>
> In practice, we can mix images generated by different attacker models to create a more robust poisoning dataset, which would help make the attack more resilient across models.
>
> ## Weakness 4
>
> Thank you for pointing out this oversight. The process for constructing prompts for generating poisoned data is outlined in Appendix 2 - Table 4. We will make this more explicit in the main body of the paper to ensure clarity.
>
> ## Weakness 5
>
> We acknowledge the discrepancy in our claim regarding the RL algorithms used with different target models. We failed to adapt RL algorithms across models appropriately. We will update the paper to clarify this issue and adjust our claims accordingly.
>
> ## Question 1
>
> Thank you for highlighting this issue. We will revise Equation 4 to use more precise notation to avoid overloading the "$\oplus$" operator, ensuring the formal definition is clearer.
>
> ## Question 2
>
> Yes, the selection of triggers requires a filtering process, as not all trigger-concept combinations will work reliably:
>
> 1. **Concept Selection**: We start by identifying concepts with attack potential, such as "black skin" to induce bias.
> 2. **Trigger Pair Matching**: For each concept, we select multiple triggers, e.g., ("attractive lady", "black skin") or ("CEO", "black skin").
> 3. **Prompt Generation**: use GPT to generate 100 diverse prompts for each trigger-concept pair.
> 4. **Concept Generation Testing**: feed these prompts into the benign target model and count how often the concept appears in the generated images.
> 5. **Stable Trigger Selection**: choose those triggers that consistently generate the target concept across different models.
>
> Regarding your second question, the "clean target model" refers to the model **before RLHF**.
>
> ## Question 3
>
> Yes, the image encoder used in the feature collision process is the same as the one used in the reward model refinement.
>
> ## Question 4
>
> As shown in Figure 6, the majority of the settings achieve good ASR performance after 800 RLHF steps. Other works on reward poisoning attacks, such as *RLHFPoison: Reward Poisoning Attack for Reinforcement Learning with Human Feedback in Large Language Models* and *Universal Jailbreak Backdoors from Poisoned Human Feedback*, do not specify the number of RLHF steps used. We chose 800 steps because, by that point, the reward values in most of our experiments stabilize.
>
> ## Question 5 & 6
>
> Yes, the training prompts are the ones paired with poisoned images.
>
> ## Question 7
>
> Apologies for the oversight. Table 2 shows the results tested on images generated by SD 3.5, under feature collision with $\beta = 0.008$, learning rate = 0.01, and max iterations = 100. We will make sure to include this clarification in the main body of the paper.
>
> ---
>
> Once again, thank you for your thoughtful feedback. We will incorporate these changes into the revised version of the paper and look forward to submitting an updated manuscript.

---

### Official Review · Reviewer_ckj4 · 2025-11-01

**Soundness:** 3
**Presentation:** 3
**Contribution:** 3
**Rating:** 4
**Confidence:** 4

**Summary:**

The paper proposes BadReward, a clean-label poisoning attack targeting the reward model used during RLHF in text-to-image diffusion model. The attack works by creating feature-collision poisoned images, which remain visually similar to benign images while being semantically aligned with target malicious concepts in CLIP embedding space. When these poisoned examples are included in preference data, the reward model learns to assign higher scores to outputs containing the target concept whenever a chosen trigger words appear.

**Strengths:**

1. The attack does not modify preference labels or require control of annotators, which is low-cost.

2. The attack pipeline and optimization objective are well explained and easy to reproduce.

3. The presentation is well and clear.

**Weaknesses:**

1. Trigger–concept selection is underspecified: The method implicitly relies on choosing trigger–concept pairs that already have some representation overlap in the model’s data distribution. This selection procedure is not formalized, and success may vary across concepts.

2. Novelty: BadReward extends existing clean-label feature-collision poisoning to the reward modeling stage rather than introducing a fundamentally new poisoning mechanism.

3. The paper does not analyze the conditions under which CLIP feature similarity reliably transfers to reward-driven policy updates.

4. The paper only tested the method on SD v1.4 or SD Turbo. More target models should be included for testing.

**Questions:**

See weaknesses.

---

> ### Author Response · Authors · 2025-11-26
>
> Thank you for your constructive feedback and insightful comments. We appreciate your time and effort in reviewing our work. Below, we address each of the weaknesses and questions you raised.
>
> ## Weakness 1
>
> We apologize for not formalizing the process of trigger-concept selection in the paper. The actual procedure is as follows:
>
> 1. Concept Selection: We begin by identifying concepts that have attack potential, such as "black skin" which can be used to induce bias.
> 2. Trigger Pair Matching: For each concept, we select k different triggers, such as ("attractive lady", "black skin") or ("CEO", "black skin").
> 3. Prompt Generation: We use GPT to generate 100 diverse descriptions for each trigger-concept pair to create prompts.
> 4. Concept Generation Testing: These prompts are then input into the original benign target model, and we count how many times the concept appears in the generated images (to ensure the attack can succeed during RLHF).
> 5. Stable Trigger Selection: We generate multiple sets of images and select those triggers that consistently produce the target concept across different models.
>
> ## Weakness 2
>
> Thank you for acknowledging that the feature collision algorithm itself does not introduce a fundamentally new poisoning mechanism.
>
> However, we believe our method offers significant value in addressing the following challenges:
>
> 1. We proposed an indirect poisoning method for Diffusion Models by making minimal modifications to the RLHF preference process.
> 2. The clean-label approach is stealthy and practical, which increases the relevance of our method in real-world scenarios.
>
> We will make sure to emphasize these contributions more clearly in the revised manuscript.
>
> ## Weakness 3
>
> Thank you for your thoughtful comment regarding the transfer of CLIP feature similarity to reward-driven policy updates.
>
> To clarify the mathematical chain of how feature similarity impacts policy updates:
> **Feature Similarity → Reward Model Bias → Policy Gradient Update**.
>
> There are two key conditions for this transfer:
>
> 1. Dependence of Reward Model on CLIP feature space: If a reward model is based on a different visual encoder, such as BLIP or DINOv2, we need to apply feature collision to the new encoder's feature space as well.
> 2. Sensitivity of RL algorithms to Reward Values: This is satisfied in PPO-based optimization strategies, where policy updates are influenced by reward model biases, making the attack feasible.
>
> These are the necessary conditions for feature similarity to transfer from CLIP to the Diffusion Policy. We will add further clarification on this in the paper.
>
> ## Weakness 4
>
> Thank you for pointing out this valuable issue. We initially encountered difficulties adapting foundational RL algorithms, such as DDPO, and state-of-the-art models based on Flow-Match (e.g., Stable Diffusion 3.5 and Flux).
>
> We have since implemented Flow-GRPO and fine-tuned it on SD 3.5, and we are in the process of applying poisoning attacks to this setup. We will add the updated ASR results to the paper as soon as possible.
>
> ---
>
> We sincerely appreciate your constructive feedback and will make the necessary revisions based on your suggestions. Thank you again for your time and valuable insights.

---

### Official Review · Reviewer_xGt9 · 2025-11-08

**Soundness:** 3
**Presentation:** 3
**Contribution:** 3
**Rating:** 6
**Confidence:** 3

**Summary:**

The paper introduces BADREWARD, a clean-label poisoning attack against reward models in text-to-image (T2I) RLHF systems. BADREWARD induces feature collisions in the CLIP embedding space to corrupt the reward model, enabling adversaries to steer T2I model outputs toward targeted malicious concepts when specific triggers are present in prompts. Notably, the attack does not require control over annotations, instead relying on stealthy manipulation of a small fraction of preference training data. Experimental evaluation on Stable Diffusion (v1.4 and Turbo) and several adversarial model generators demonstrates high attack success rates, notable stealthiness by visual and perceptual metrics.

**Strengths:**

1. The black-box scenario assumes the adversary can inject even a small fraction of poisoned pairs into the RLHF pipeline. In many real-world alignment pipelines, this step is subject to curation, filtering, or annotation review, which could detect subtle distribution shifts. The paper does not investigate the sensitivity of system-level detection to these injections.
2. While Table 2 and Figure 4 demonstrate strong SSIM/LPIPS results, stealth evaluation is reduced to pixel-level or shallow perceptual metrics. The paper does not assess detectability by automated anomaly detection methods or statistical audit systems that could flag poisoned distributions in embedding or reward space.
3. Reward models are built on a fixed CLIP backbone with a simple MLP. There is little discussion of how architecture choices, frozen vs. trainable backbone, or reward model complexity affect the attack’s transferability/durability.

**Weaknesses:**

1. Can the authors provide quantitative comparisons with recent SOTA RLHF poisoning attacks in terms of both effectiveness and detectability?
2. Have the authors evaluated whether feature-collided samples can be detected by statistical anomaly methods operating in embedding, reward, or preference score space, beyond pixel-perceptual similarity metrics?
3. Does BADREWARD generalize to subtler forms of steering (e.g., more abstract style or concept changes), or is it reliant on visually salient feature collisions?
4. Can the authors provide more insight or results on how different choices of the $\beta$ parameter and initial images ($x_b$, $x_t$) affect stealth and ASR?

**Questions:**

Refer to the weaknesses.

---

> ### Author Response · Authors · 2025-11-26
>
> Thank you for your thoughtful and detailed feedback. We appreciate your insights and suggestions, and we address each of the weaknesses and questions raised below.
>
> ## Weakness 1
>
> In our research, we found few other work has targeted Reward Poisoning attacks during the RLHF process of Diffusion Models. Similar works, such as *Backdooring Bias ($B^2$) into Stable Diffusion Models*, focus on the SFT stage, while some RLHF poisoning attacks for LLMs, such as *RLHFPoison: Reward Poisoning Attack for Reinforcement Learning with Human Feedback in Large Language Models*, are more similar to our semantic-level poisoning method.
>
> To compare the effectiveness of semantic-level and feature-level attacks, we conducted an attack success rate (ASR) evaluation across different settings:
>
> | Attacker Model - Target Model | Attack Goal               | ASR-Semantic-level | ASR-Feature-level |
> | ----------------------------- | ------------------------- | -------------- | ------------- |
> | SD v3.5 - SD v1.4             | (Old, Eyeglasses)         | 0.97           | 0.98          |
> | SD v3.5 - SD v1.4             | (Attractive, Black Skin)  | 0.94           | 0.89          |
> | SD v3.5 - SD v1.4             | (Traffic Accident, Blood) | 0.95           | 0.84          |
> | Cogview4 - SD v1.4            | (Old, Eyeglasses)         | 0.93           | 0.83          |
> | Cogview4 - SD v1.4            | (Attractive, Black Skin)  | 1.00           | 0.92          |
> | Cogview4 - SD v1.4            | (Traffic Accident, Blood) | 0.92           | 0.86          |
>
> The results show that, in most settings, the feature-level attack has a slightly lower ASR compared to the semantic-level approach, which directly flips labels. However, the trade-off between lower ASR and the clean-label stealthiness of our method makes it highly practical.
>
> ## Weakness 2
>
> Thank you for pointing this out. We recognize that feature collision in the CLIP embedding space can be detected through distribution shifts. We will update the "Possible Countermeasures" section in the revised version of the paper to include this analysis and further explore potential defenses.
>
> ## Weakness 3
>
> We appreciate your observation and it's quite feasible to generalize. Similar issues were observed in Appendix D, Figure 7, where the attack target was to generate images of an old person wearing glasses when the prompt contained "old." However, the adversarial model, Cogview4, produced many comic-style images, which led to the RLHF-tuned SD v1.4 model generating similar styles. This phenomenon is worth exploring, and we plan to include additional experiments to investigate how different attack goals and styles may affect the results.
>
> ## Weakness 4
>
> The $\beta$ parameter in the feature collision optimization controls how far the optimized image $x$ can deviate from the benign image $x_b$. Below are the results for SD v3.5 generating images with the attack goal (old, eyeglasses) for different values of $\beta$, showing the trade-off between stealth (PSNR) and reward model effectiveness (Reward Overlap):
>
> | $\beta$ | lr   | PSNR $\uparrow$ | Reward Overlap $\uparrow$ |
> | ------- | ---- | --------------- | ------------------------- |
> | 0.005   | 0.01 | 24.97 dB        | 0.945                     |
> | 0.008   | 0.01 | 27.70 dB        | 0.904                     |
> | 0.01    | 0.01 | 28.10 dB        | 0.889                     |
>
> Regarding the choice of ($x_b$, $x_t$), it is difficult to quantify in a single metric. Intuitively, if the embedding difference between $x_b$ and $x_t$ in CLIP is larger, the feature-collided sample will have better separation in the trained reward model, which may indirectly improve the ASR. We plan to include further exploration of this in our updated paper.
>
> ---
>
> Thank you once again for your valuable feedback. We will work on incorporating your suggestions into the next version of the paper and look forward to sharing the updated results with you.

---

### Official Review · Reviewer_6cEx · 2025-11-09

**Soundness:** 2
**Presentation:** 4
**Contribution:** 3
**Rating:** 4
**Confidence:** 4

**Summary:**

This work introduces a method for compromising the reward model used in the post-training of T2I systems. The attack injects explicit and adversarially crafted poisoned training examples, causing the T2I model to generate harmful or inappropriate content upon encountering specific trigger prompts.

**Strengths:**

Reward-model poisoning in T2I RLHF is underexplored relative to SFT-time poisoning; the paper clearly motivates why RLHF is a sensitive surface.

**Weaknesses:**

1. **Reward-model diversity** The **Feature-Level Poisoning Attack** is evaluated in a CLIP-on-CLIP setting: poisons are crafted with **white-box** access to a CLIP encoder and tested on CLIP-based reward models, which lowers the difficulty and obscures generality and makes the gray-box and black-box threat model overclaimed. Please evaluate on non-CLIP reward backbones (e.g., BLIP, HPSv2, ImageReward), test reward ensembles and multi-reward optimization/confidence-aware training, and report ASR to show the effectiveness of the Feature-level Poisoning Attack.

2. **Lack of evaluation under basic defenses.** Missing evaluation against basic safety defenses. The RLHF setup optimizes semantic/aesthetic alignment rather than safety, so the attacked T2I models are effectively unguarded. In practice, providers deploy baseline defenses—harmful-concept erasure, safety-focused RLHF, and runtime safety filters (e.g., the SD-1.5 Safety Checker, Q16). Please evaluate BadReward under these defenses (individually and combined) and report ASR and utility on benign prompts. If standard guardrails block the attack, its practical impact is limited; if not, show how the method bypasses them.

3. **less informative citations** The claim in L39 that “RLHF is an indispensable component for aligning T2I systems with human expectations” is unsupported. Please provide primary evidence—e.g., technical reports or production case studies from major T2I providers, or peer-reviewed ablations—demonstrating indispensability. Likewise, using broad surveys (e.g., Zhu et al., 2024 at L42) to justify specific limitations of RLHF is insufficient; cite targeted primary sources that document these limitations in T2I settings. If such evidence is unavailable, soften the claim (e.g., “commonly used” or “increasingly adopted”) and reposition the motivation accordingly.

**Questions:**

Please refer to the weakness section and:

Experimental setup — missing details


1.**Reward-model training**
Specify what is updated: full backbone vs. MLP head only. Clarify any frozen layers, parameter count, and basic optimizer settings (lr, epochs, regularization).

2. **Poison budget**
Report the number and percentage of semantic-level poisoned pairs used for reward training, plus the mixing schedule per epoch (and whether poisons appear in validation).

3. **Trigger coverage**
Why only three triggers? Please report ASR as the number and diversity of triggers increase (including synonyms/paraphrases/typos) and provide variance across seeds to show robustness.


4. What is the minimum effective poison budget across targets and training seeds for stable success (1%, 2%, 3% are shown)? Can you report variance across seeds?

---

> ### Author Response · Authors · 2025-11-26
>
> Thank you for your detailed and constructive feedback. We greatly appreciate your comments and suggestions, and we will address each of the weaknesses and questions raised.
>
> ## Weakness 1
>
> We conducted a simple case study to evaluate the feature collision attack across three visual encoders: CLIP, BLIP2, and DINOv2. We used 384 poisoned pairs (3% poison ratio) and calculated MSE loss between the base image ($x_b$) and the poisoned image ($x$), as well as between the target image ($x_t$) and the poisoned image ($x$). The results are as follows:
>
> | Encoder | MSE($x_b$, x) | MSE($x_t$, x) |
> | ------- | ------------- | ------------- |
> | CLIP    | 8.09e-4       | **3.82e-5**   |
> | BLIP2   | 5.41e-4       | 2.42e-4       |
> | DINOv2  | 9.24e-4       | 3.13e-4       |
>
> The results show that while the feature collision samples are still close to the target $x_t$ in other encoders, the proximity is not as strong as in the CLIP setting. We further trained a BLIP2-based reward model using poisoned data from the CLIP-based feature collision attack, but it was not as effective at rewarding malicious concepts as the CLIP-based reward model:
>
> | Reward Score | w/o Malicious Concept | with Malicious Concept |
> | ------------ | --------------------- | ---------------------- |
> | CLIP-based   | 0.013                 | 0.998                  |
> | BLIP2-based  | 0.117                 | 0.404                  |
>
> As a next step, we are exploring a method based on Canonical Correlation Analysis (CCA) to identify a common subspace across different image encoders. For example, we computed the lower-dimensional shared feature space between CLIP and BLIP2 for "traffic accident" images using CCA, then performed feature collision in this space. Preliminary results show promising outcomes for reward models based on both CLIP and BLIP2:
>
> | Reward Score | w/o Malicious Concept | with Malicious Concept |
> | ------------ | --------------------- | ---------------------- |
> | CLIP-based   | 0.020                 | 0.856                  |
> | BLIP2-based  | 0.049                 | 0.794                  |
>
> Although the reward values are slightly lower than the "CLIP-to-CLIP" setting, the results are still promising in guiding malicious concept rewards.
>
> We are working to complete RLHF experiments and will update the PDF with results and details of the CCA algorithm soon.
>
> ## Weakness 2
>
> ### 1. Runtime Safety Checker
>
> We downloaded the `CompVis/stable-diffusion-safety-checker` from HuggingFace and used it to classify generated images as "safe" or "unsafe." The results are as follows:
>
> | Concept $\mathcal{C}$ | Total Number | Safe | Unsafe |
> | --------------------- | ------------ | ---- | ------ |
> | Blood on the ground   | 256          | 241  | 15     |
> | Eyeglasses            | 256          | 217  | 39     |
> | Black Skin            | 256          | 227  | 29     |
>
> The results show that the Stability Diffusion Safety Checker has limited ability to handle the harmful concepts defined in our experiments.
>
> ### 2. Harmful-Concept Detection/Erasure
>
> For detecting feature distribution shifts due to poisoning across the dataset, we agree that clustering methods can be employed to identify anomalies. We are working on additional experiments and will update the "Possible Countermeasures" section in the revised paper.
>
> ## Weakness 3
>
> We apologize for the inappropriate wording in the paper. We will soften the claim in the revised version. Thank you for pointing this out!
>
> ## Questions
>
> 1. Only the MLP head is updated during training. We train for 20 epochs, with a learning rate of 1e-3 for the first 10 epochs and 1e-4 for the last 10 epochs. We use the AdamW optimizer for training.
> 2. The clean dataset contains 13,000 image pairs. For the 1%, 2%, and 3% poison ratios, we use 128, 256, and 384 poisoned pairs, respectively.
> 3. Thank you for your suggestion. We are conducting more experiments with a wider range of trigger-concept pairs, such as (criminal-black skin) and (photo of cat, cartoon style). We will include these results in the revised paper.
> 4. We did not control the seeds during the reward model training. The seeds for RLHF (DDPO, SDPO) are set to seed=42 for the results shown in the paper. We are conducting experiments with additional seeds and will update the paper with these results soon.
>
> Thank you again for your valuable feedback. We look forward to submitting an improved version of our paper that addresses your concerns.

---

### Note · Authors · 2025-11-28

I have read and agree with the venue's withdrawal policy on behalf of myself and my co-authors.